# 大作业 **Proposal** 说明

【前情回顾】大作业占总成绩 60%，2-3 名学生组成一个团队，应用机器学习解决一个实际问题。可以自行选与大模型相关的主题（推荐）或者个人比较感兴趣的机器学习主题，也可以选择 Kaggle 上的一个任务（http://www.kaggle.com/competitions）。我们希望在你的最终报告上看到以下内容：问题（是什么）、动机（为什么）、技术（怎么做）、结果与分析（你验证了上面声称的内容吗）和结论。需要提交的材料包括：一份 Proposal（第 6 周周日）、一份期中报告和 PPT（第 9 周周三上课前）、一份最终报告、PPT 和实现代码（第 17 周周三上课前提交 PPT，周日提交其他内容）。除去 PPT 外的所有报告材料均需采用 NIPS 格式，并用英语书写，样式文件可在此获取：

https://neurips.cc/Conferences/2023/PaperInformation/StyleFiles。第 11 周进行 Proposal 和中期演示，第 17 周进行海报演示。

【本次说明】请严格按照下述要求完成全部指定任务，Proposal 在最终的成绩中占比 10%，中期 20%，结题 30%，共 60%。每一次成绩都会以百分制返回，比如 Proposal 最终返回一个 100 以内的数字，在最终计算成绩的时候会*10%，比如你得 60 分 Proposal，那么最终总成绩里面 proposal 就占 6 分，依次类推。下面的说明也是按照 100 分进行分配的。

【1】注册 openreview 平台（5 分）
openreview 注册流程教程 https://blog.csdn.net/YP_67689E4F/article/details/138245504

【2】补充完整选题信息和队伍信息，特别是邮箱，否则后面无法互评：（5 分）
位置：https://zhipu-ai.feishu.cn/sheets/ZelLspHPYhbDc3tCECacWSSSno6?from=from_copylink

【3】助教会在统计到大家的邮箱后将各位同学设置为审稿人，请关注相关"网络学堂"通知和"课程群"助教通知，配合完成正确操作（5 分）

【4】proposal 提交到"网络学堂"（5 分）
具体 Proposal 要求见后文，注意同一选题只需要提交一次，注意按照贡献标明作者排序，需包含选题内所有同学

【5】proposal 提交到"openreview"平台（5 分）
提交要求：命名为"【Proposal】*******"，必须包含【Proposal】，否则视为无效提交
注意同一选题只需要提交一次，注意按照贡献标明作者排序，需包含选题内所有同学
提交地址：场地主页：https://openreview.net/group?id=tsinghua.edu.cn/THU/2024/Fall/AML

【6】Proposal 互评（25 分，预计由 5 人及以上评每个 proposal，每人平均 5 分）
互评方式预计使用 openreview，如果存在问题会采用网络学堂，请关注相关"网络学堂"通知和"课程群"助教通知，配合完成正确操作

【7】Proposal 助教评和教师评（50 分，三位助教各 10 分，教师 20 分）

补充说明：如果你有算力需求以及下文提到的接口调用需求，请及时在网络学堂中发布的"算力需求"问卷中填写清楚，比如如果你愿意尝试，请在 https://open.bigmodel.cn/ 注册你的个

人账户。我们将提供免费 tokens，你需要在问卷中提供你的用户名以获得免费预算。

## Project Proposal

Advanced Machine Learning, 2024 Fall

**Due: 2024-10-20 11.59 PM.** You should submit a **PDF** format proposal to **Tsinghua Web Learning and Openview** written by LATEX in **NeurIPS conference paper format** (which is inline with the final report's requirement. Visit NeurIPS official website or Overleaf for templates). The proposal should be no longer than **2 pages (excluding references)** and every team (2-4 students) only needs to submit one proposal. The proposal can be in either Chinese or English.

## 1 Requirements

In this assignment, you should propose a research proposal based on Advanced Machine Learning. You need to develop new machine learning methods for established problems or newly defined topics. Specifically, the following suggestions may be helpful to your proposal writing:

1. **Background**: What's the background of the problem? Is it theory-driven, or deeply rooted in some useful application situations? Is it important or necessary? What impact will it bring if you finally solve it?

2. **Definition**: Is there any formal or mathematical definition for your problem? Explain the symbols you may want to use in the proposal writing.

3. **Related Work**: Is your problem a well-established one? If so, review existing approaches and discuss their advantages and disadvantages; if not, survey and describe related problems in the field and list some potentially applicable baseline methods. Remember to cite related work properly.

4. **Proposed Method**: What are the motivations for you to choose it? Which datasets do you propose to experiment on? What baseline approaches do you plan to compare with? How do you implement your proposed method based on the dataset? It is ok to change and improve it later but now try to describe it as detailed as possible.

Additionally, the proposal PDF should include your team member's name(s) and student ID(s). **Plagiarism is strictly prohibited**, or you will fail the course.

## 2 Examples and Resources

Some potential directions for your reference:

1. **Alignment, particularly Reinforcement Learning-based Alignment**: In existing research, the application of reinforcement learning methods to alignment tasks has sparked wide discussions. However, these works usually make only slight adjustments to the loss function, and their experimental results have not shown significant improvements. Therefore, exploring more effective reward models becomes a key issue. Research directions can include defining what constitutes a "good reward" and how to provide more fine-grained and accurate rewards, rather than merely focusing on the dimensions of usefulness and toxicity.

2. **Long Text Modeling**: Long text modeling is currently a core opportunity for the

commercialization of large language models (LLMs), but this field has not yet reached an optimal level. Research can seek breakthroughs in methodology, such as optimizing attention mechanisms and ROPE (Relative Position Encoding). Furthermore, there are many unresolved issues regarding long text modeling of Retrieval-Augmented Generation (RAG). Both academia and industry lack sufficient research and benchmarking in this field, making it a promising area for development.

3. **Retrieval-Augmented Generation (RAG)**: Retrieval-Augmented Generation (RAG) improves AI system performance by combining information retrieval and text generation. Its core advantage is the use of external knowledge bases to assist the generation process, enhancing the accuracy and robustness of the content. Research focuses include efficiently integrating retrievers with generators, exploring cross-modal applications, and knowledge updates. Current challenges are mainly concentrated on improving retrieval efficiency, enhancing generation quality, and achieving cross-domain applications.

4. **Agents**: With the extensive use of large models in complex application scenarios, building intelligent agents relying on large models has become a trend. For instance, Microsoft researchers have explored the importance of Agent AI, particularly in the aspects of physical, virtual reality, and sensory interactions. Research directions include multi-task learning, commonsense reasoning, and continual learning, aiming to improve the performance and adaptability of agents across various tasks.

5. **Mamba**: Mamba is a Selective Structured State Space Model that excels in linear-time reasoning, parallel training, and robust performance for long-context tasks. Proposed by CMU, this model alleviates the limitations of convolutional neural networks through global receptive fields and dynamic weighting, while providing advanced modeling capabilities similar to Transformers but avoiding the computational complexity associated with them. Research directions focus on enhancing the ability to process long-sequence data, multi-modal data handling, and computational efficiency.

6. **MoE (Mixture of Experts)**: MoE, or Mixture of Experts Models, was initially proposed by researchers at the University of Cambridge in 1991. However, with the development of large-scale models and multi-task applications, MoE has once again become a research hotspot. Exploring the applications of MoE in current large model frameworks, especially in dealing with complex tasks and improving model efficiency, is of significant research value.

7. **Training or Fine-tuning Image Understanding Based on Pre-trained Language Models**: With the rapid development of large-scale pre-trained text models, researchers have gradually extended pre-training to more modalities. One subfield is image understanding, including tasks such as image question answering and image caption generation. A common approach is to align images to the text feature space to fully leverage pre-trained text models. Thus, research can focus on how to effectively align images with text features to improve the performance of image understanding tasks.

8. **Research on Recommendation Algorithms Based on Large Language Models (LLM)**: Due to their excellent language understanding, reasoning, and planning capabilities, large language models have shown great potential in various fields, driving new developments in information retrieval and recommendation systems. Key techniques include triggering the recommendation capabilities of LLMs through context learning and aligning LLMs for recommendation tasks based on instruction fine-tuning. Students can choose a subfield

related to their life or research and explore applying pre-trained language models to improve the performance of recommendation systems.

9.  **Fine-tuning and Applications of Open-Source Image Generation and Diffusion Models**: Diffusion models are common pre-trained image generation models and have performed excellently in research and application fields. Training a text-to-image generation diffusion model from scratch is extremely costly; however, open-source models (such as Stable Diffusion) offer the opportunity to fine-tune pre-trained model parameters to endow the model with new capabilities or further enhance performance in certain aspects. Research can focus on how to effectively fine-tune these models to achieve task-specific improvements.

10. **Applications of Large Language Models (LLMs)**: In recent years, the rise of LLMs such as ChatGPT has garnered widespread attention. Unlike traditional end-to-end supervised learning, most applications created by LLMs are based on zero-shot or few-shot learning via APIs and do not require local GPUs. If ample resources are available, attempting to develop other LLM applications or improve LLMs, such as through acceleration and fine-tuning, can be explored.

11. **Kaggle or Other Public Competitions**: Kaggle offers many datasets and established benchmarks, though their quality may vary. The competition tracks at NeurIPS 2022 are also a challenging option. These competitions provide students with opportunities to tackle real-world problems and solutions, helping them enhance their skills through practical applications.

12. **Publishing in Journals or Conferences**: Track some recently published machine learning papers (e.g., from ICML, NeurIPS, ICLR, KDD, ACL, CVPR). These papers usually provide detailed datasets and benchmark results. Students can attempt to replicate these papers' results and then try to make improvements to add value to their final project.