# OpenReview forum: "【AML】Project Proposal Guidelines for Advanced Machine Learning Course"
_tsinghua.edu.cn/THU/2024/Fall/AML — THU 2024 Fall AML Submission_

### Official Review · ~Zijun_Liu2 · 2024-11-06
**Thanks for the Guidelines**

**Rating:** 9
**Confidence:** 4

**Review:**

The guidelines for the proposal is clear and informative. However, it will be better to include guidelines for reviewers, e.g., what should be included in a review. Thank TAs for the effort again.

---

### Official Review · ~Daniel_Wang4 · 2024-11-06
**Clear Guidelines**

**Rating:** 9
**Confidence:** 4

**Review:**

The AML project guideline is well-structured, covering all submission stages and using the OpenReview platform. Includinga reviewer guideline would help us understand clearly what we are expected to do.
Overall, the guideline is very clear.

---

### Official Review · ~Peidong_Zhang1 · 2024-11-08
**Extremely helpful**

**Rating:** 7
**Confidence:** 4

**Review:**

The guideline is very clear and helps a lot.

---

### Official Review · ~Aleksandr_Algazinov1 · 2024-11-09
**Clear and helpful**

**Rating:** 10
**Confidence:** 4

**Review:**

Thank you for well-written and well-structured guidelines!

---

### Official Review · ~Yanchen_Wu1 · 2024-11-09
**Helpful guidelines**

**Rating:** 10
**Confidence:** 5

**Review:**

This guideline will help us complete the final project of the course!

---

### Official Review · ~Anton_Johansson1 · 2024-11-09
**Helpful and clear guidelines**

**Rating:** 9
**Confidence:** 4

**Review:**

Good and well-structured guidelines, thanks!

---

### Official Review · ~Tim_Bakkenes1 · 2024-11-09
**Clear guidelines**

**Rating:** 9
**Confidence:** 4

**Review:**

The guidelines are very clear, and they helped us greatly in understanding how to write the proposal. The examples provided were also very helpful in finding an interesting project.

---

### Official Review · ~Lei_Wu17 · 2024-11-09
**Perfect!**

**Rating:** 10
**Confidence:** 5

**Review:**

I think this explanation for proposal is complete and understandable enough.

---

### Official Review · ~Yunghwei_Lai1 · 2024-11-10

**Rating:** 10
**Confidence:** 5

**Review:**

Well structured guidelines, thanks for the information.

---

### Official Review · ~Jia-Nuo_Liew1 · 2024-11-10
**Clear guidelines**

**Rating:** 10
**Confidence:** 5

**Review:**

Well structured, straightforward and clear guidelines for the proposal.

---

### Official Review · ~Xun_Wang10 · 2024-11-10
**GOOD GUIDELINES**

**Rating:** 10
**Confidence:** 5

**Review:**

It is a clear guidelines.

---

### Official Review · ~Rosalie_Butte1 · 2024-11-10
**clear guidelines**

**Rating:** 10
**Confidence:** 5

**Review:**

Clear and well-structured guidelines!

---

### Official Review · ~Chentian_wei1 · 2024-11-10
**clear**

**Rating:** 10
**Confidence:** 5

**Review:**

clear

---

### Official Review · ~Tianxing_Yang1 · 2024-11-10
**Clear Guideline**

**Rating:** 10
**Confidence:** 4

**Review:**

It's a clear guideline for students.

---

### Official Review · ~Matteo_Jiahao_Chen1 · 2024-11-10
**Clear guidelines**

**Rating:** 9
**Confidence:** 4

**Review:**

Well-structured and clear guidelines

---

### Official Review · ~Hector_Rodriguez_Rodriguez1 · 2024-11-11
**Clear Project Proposal Guidelines but Could Use More Detail**

**Rating:** 8
**Confidence:** 4

**Review:**

The project proposal guidelines are clear and provide a good foundation to write the project proposal. However, they could benefit from including a more detailed explanation of the evaluation criteria. For example, a checklist could be added to standarize the peer review process.

---

### Official Review · ~Gausse_Mael_DONGMO_KENFACK1 · 2024-11-11
**Clear Guidelines**

**Rating:** 9
**Confidence:** 4

**Review:**

It provides clear guidelines with relevant details, explains the exercise well, and includes examples that are especially helpful for those who may not have a specific subject in mind.

---

### Official Review · ~Jin_Zhu_Xu1 · 2024-11-11
**Well explained**

**Rating:** 10
**Confidence:** 4

**Review:**

The guidelines are clear enough

---

### Official Review · ~Jiajun_Xu3 · 2024-11-11
**Helpful Guidelines**

**Rating:** 9
**Confidence:** 4

**Review:**

It's very nice of the TA to provide such clear and well constructed guidelines, helping us understand the requirements of this project.

---

### Official Review · ~Huajun_Bai1 · 2024-11-11
**Thanks**

**Rating:** 10
**Confidence:** 5

**Review:**

The guidelines are clear enough

---

### Official Review · ~Changsong_Lei2 · 2024-11-11
**Clear Guidelines**

**Rating:** 10
**Confidence:** 5

**Review:**

The guideline is clear enough for me to understand AML, very helpful!

---

### Official Review · ~Michael_Hua_Wang1 · 2024-11-11

**Rating:** 8
**Confidence:** 4

**Review:**

The guidelines are very helpful with respect to identifying a project to work on and to how to write the proposal. However, they are somewhat ambiguous with respect to what the expectations are with respect to the contents of each review.

---

### Official Review · ~Jiaxiang_Liu7 · 2024-11-11
**Thanx**

**Rating:** 10
**Confidence:** 5

**Review:**

Exceptionally clear.

---

### Official Review · ~XueZeng1 · 2024-11-11

**Rating:** 10
**Confidence:** 5

**Review:**

Clear and helpful

---

### Official Review · ~KAI_JUN_TEH1 · 2024-11-11
**A perfect handbook of guidelines**

**Rating:** 10
**Confidence:** 5

**Review:**

The guidelines are very clear and have been very helpful for my research progress as well as the writing of my proposal.

---

### Official Review · ~Ruitao_Jing1 · 2024-11-12
**A Nice Guideline**

**Rating:** 10
**Confidence:** 5

**Review:**

Very specific, clear and helpful.

---

### Official Review · ~Eddy_Yue1 · 2024-11-12
**Good**

**Rating:** 9
**Confidence:** 4

**Review:**

Easy to understand

---

### Official Review · ~Zou_Dongchen1 · 2024-11-12
**Got the information**

**Rating:** 10
**Confidence:** 5

**Review:**

Clear guidelines, thanks for the information.

---

### Official Review · ~Mingdao_Liu1 · 2024-11-12

**Rating:** 10
**Confidence:** 5

**Review:**

Awesome guideline.
~~(But why this is assigned?)~~

---

### Official Review · ~Zihan_Wang7 · 2024-11-12
**Clear and bilingual**

**Rating:** 9
**Confidence:** 5

**Review:**

Summarize:
The enclosed document serves as a comprehensive instructional manual on the formulation of a proposal.

Summary Of Strengths:
Clear and bilingual.

Summary Of Weaknesses:
Inconsistent typography between Chinese and English, cloud have better page breaks.

---

### Official Review · ~Wanlan_Ren1 · 2024-11-12
**Great guidelines!**

**Rating:** 10
**Confidence:** 5

**Review:**

This guideline is helpful and clear.

---

### Official Review · ~Kairong_Luo1 · 2024-11-12
**Great Guideline!**

**Rating:** 10
**Confidence:** 5

**Review:**

1. The topic list is inspiring and insightful;
2. The requirement is very clean;
3. The instructions are easy to follow.

---

### Official Review · ~Han-Xi_Zhu1 · 2024-11-12
**Great Guideline for the Course**

**Rating:** 10
**Confidence:** 5

**Review:**

This work briefly introduced the guideline of the AML course. The layout is clear and easy to follow. The paper is well organized. Thank you!

---

### Official Review · ~Justinas_Jučas3 · 2024-11-12
**Great work!**

**Rating:** 10
**Confidence:** 5

**Review:**

State of the art guidelines!

---

### Official Review · ~Gangxin_Xu1 · 2024-11-12

**Rating:** 10
**Confidence:** 5

**Review:**

? but !

---

### Official Review · ~Kittaphot_Saengprachathanarak1 · 2024-11-12
**Well-made and detail guidelines**

**Rating:** 10
**Confidence:** 5

**Review:**

Provide detailed description of every part in the paper and good examples!

---

### Official Review · ~jin_wang30 · 2024-11-12
**Wonderful guidance**

**Rating:** 10
**Confidence:** 5

**Review:**

Very specific, clear and helpful.

---

### Official Review · ~Fei_Long3 · 2024-11-12
**Wonderful Guideline!**

**Rating:** 10
**Confidence:** 5

**Review:**

This guideline is exceptionally well-crafted, offering clear guidance and insightful explanations that make clear guidance accessible to learners!

---

### Official Review · ~Zhu_Zhang6 · 2024-11-12
**Excellent guideline**

**Rating:** 10
**Confidence:** 5

**Review:**

This is an excellent guideline, providing a clear understanding of the tasks to be completed and highlighting the key areas of focus in the AML course. We greatly appreciate the teaching assistants' patience and dedication in designing and organizing the course assignments, which enhances our learning experience and helps us to grasp the essential aspects of the curriculum more effectively. Thank you for all the effort and support!

---

### Official Review · ~Grace_Xin-Yue_Yi1 · 2024-11-12

**Rating:** 10
**Confidence:** 5

**Review:**

Clear and helpful guidelines

---

### Official Review · ~Yifan_Luo2 · 2024-11-12
**Good**

**Rating:** 10
**Confidence:** 5

**Review:**

Good

---

### Decision · Program_Chairs · 2024-11-18

**Decision:**

Best Paper

**Comment:**

**Advantages:**
1. Clear task objectives
2. Provides guidance
3. Plays a significant role in selecting research topics

**Disadvantages:**
1. Inconsistent formatting between Chinese and English
2. Needs more specific research objectives for each topic
3. Lacks relevant guidelines for reviewers